

# Cloud optical and physical properties retrieval from EarthCARE multi-spectral imager: the M-COP products

Anja Hünerbein[1], Sebastian Bley[1], Hartwig Deneke[1], Jan Fokke Meirink[2], Gerd-Jan van Zadelhoff[2], and Andi Walther[3]

[1]Leibniz Institute for Tropospheric Research, Leipzig, Germany
[2]Royal Netherlands Meteorological Institute, De Bilt, The Netherlands
[3]Cooperative Institute for Meteorological Satellite Studies, University of Wisconsin-Madison, Madison, United States

**Correspondence:** Anja Hünerbein (anjah@tropos.de)

**Abstract.** ESA's Cloud, Aerosol and Radiation Explorer EarthCARE is the first mission which will provide measurements from active profiling, passive imaging and a broad-band radiometer from a single satellite platform. The passive multi-spectral imager (MSI) features four solar and three thermal infrared channels, and has a swath of 150 km and a spatial pixel resolution of 500 m. The MSI observations will provide across-track information on clouds and aerosol to extend the active profiling

information into the swath. In this paper, we present the algorithm used for retrieving the cloud optical and physical products (M-COP), specifically cloud optical thickness, effective radius and top height. The algorithm is based on the solar and terrestrial MSI channels within an optimal estimation framework. The advantage of optimal estimation is that it enables full error propagation given by the uncertainties in measurements and a-priori information. The MSI cloud algorithm has been successfully exercised on different imagers and on synthetically generated MSI observations.

## 10   1   Introduction

Clouds play an important role in the Earth's radiation budget due to their strong interaction with atmospheric radiation, and the resulting complex influence on radiative fluxes (Hartmann and Short, 1980). Global-scale satellite observations of clouds and radiation are thus essential for our knowledge about the Earth's energy budget and climate system. Of particular scientific interest are cloud changes in response to anthropogenic influences, including aerosol emissions, which manifest themselves

as radiative forcing and can potentially contribute to a number of feedback processes and thus further amplify climate change (IPCC 2021, in press).

Despite recent progress in constraining the uncertainty of the overall cloud feedback – estimated to have a positive value of $0.42\,Wm^{-2}$ in the Sixth Assessment Report of the International Panel on Climate Change (IPCC 2021, in press), there is an urgent need for long-term, high-quality global-scale observations to improve our understanding of cloud processes including

their representation in weather and climate models. Here, the interaction of aerosol particles and clouds deserves particular focus, due to the uncertainty introduced in climate change projects.

Since the early 1980's, a number of satellite missions have provided multi-spectral imagery as basis for long-term global-scale cloud and radiation data sets. A particularly noteworthy effort is the International Satellite Cloud Climatology Project



(ISCCP), which utilized both the early geostationary satellite instruments and the Advanced Very High Resolution Radiometer
(AVHRR) flown on NOAA's operational polar-orbiting satellite series. The inter-calibration of individual sensors to yield a homogeneous record was found to deserve special attention for data consistency. Over time, the observational capabilities of these passive imaging spectrometers have also greatly improved both for polar-orbiting (MODIS, VIIRS) and geostationary satellites (GOES, MSG, HIMAWARI).

A variety of methods exist for the determination of cloud microphysical properties from multi-spectral satellite imagery,
for example from AVHRR (Derrien et al. (1993); Kriebel et al. (1989); Nakajima and King (1990); Walther and Heidinger (2012)), ATSR (Watts et al., 1998), MODIS (King et al., 1997), GOES and Meteosat (Minnis and Harrison, 1984). A baseline of all these techniques is the fact that the reflection of clouds at a non-absorbing channel in the visible wavelength region is primarily dependent on the cloud optical thickness; whereas the reflection function at a water- or ice-absorbing wavelength in the shortwave-infrared region is primarily a function of cloud particle size or, equivalently, single scattering albedo.

Hence, at present, there are several long-term, consistent satellite-based climate data records of cloud properties available from passive satellite imagery, with various differences in time coverage, accuracy and availability of parameters, which ultimately determines their suitability for a particular scientific question (Stubenrauch et al., 2013).

Passive multi-spectral satellite imagers are, however, fundamentally limited by their measurements' principle, which provides only limited and indirect information on cloud vertical structure. In this respect, the launch of the Cloudsat and CALIPSO
(Cloud and Aerosol Lidar and Infrared Pathfinder Mission) satellites in 2006 to join the A-Train satellite constellation opened up a new perspective from space, revealing for the first-time detailed information about the complex vertical structure of cloud and aerosol profiles. The two satellites carried the Cloud Profiling Radar (CPR) and the CALIOP (Cloud and Aerosol Lidar with Orthogonal Polarization) instruments, respectively, as sources of the first long-term active remote sensing observations from space (Stephens and Kummerow, 2007). A drawback of such active observations is the restriction of coverage to a rather
narrow region along the satellite ground track. In addition, flying instruments on separate satellite platforms makes the subsequent data analysis susceptible to temporal changes in atmospheric state. While it was possible to recover Cloudsat operations after a spacecraft anomaly, tight collocation with Calipso and MODIS could no longer be guaranteed, making it impossible to produce the synergistic Cloudsat, Calipso and MODIS product with sufficient accuracy (Kato et al., 2010).

The EarthCARE satellite mission is a joint mission by the European and Japanese Space Agencies, and is ESA's 6th Earth
Explorer. The spacecraft will feature four instruments on a single platform: the Cloud Profiling Radar (CPR) with Doppler capability, the high spectral resolution Atmospheric Lidar (ATLID), the Multi-Spectral Imager (MSI), and the Broad-Band Radiometer (BBR). The satellite measurements will be used to retrieve global profiles of cloud, aerosol, and precipitation properties and along with top of atmosphere terrestrial and solar fluxes (Illingworth et al., 2015). The MSI instrument is measuring in 7 channels in the visible, near infrared and infrared spectrum, with central wavelengths of 0.67 (VIS), 0.865
(NIR), 1.65 (SWIR1), 2.21 (SWIR2), 8.8 (TIR1), 10.8 (TIR2), and 12.0 $\mu$m (TIR3), and with 500 m spatial resolution (Wehr et al., 2023). MSI will have a swath width of 150 km, asymmetrically tilted away from the sun and covering -35 km to 115 km relative to nadir. MSI observations will be used to provide cloud products and to extend the spatially limited coverage of cloud properties obtained from the active sensors into the across-track direction. Furthermore, the information gained from the



MSI swath will make it possible to identify weather phenomena at the mesoscale or even synoptic scale, and utilize these as
context for the interpretation of the active observations. The cloud micro-physical retrievals are based on the combination of the
non-absorbing visible channels and the absorbing near infrared channels. MSI has no dedicated absorption channels for cloud
top determination. Therefore, the cloud top height retrieval is limited to the information from the infrared window-channels
(Inoue (1985), Fritz and Winston (1962)). To compensate for the instrument limitations and ensure consistency between the
products, the cloud microphysical retrieval is combined with the macro-physical retrievals into one retrieval algorithm similar
to Watts et al. (2011) or Poulsen et al. (2012).

The goal of the present paper is to introduce the algorithms used as basis for the cloud property retrievals, which will
be applied operationally to MSI observations as part of the MSI cloud processor (M-CLD). The cloud optical and physi-
cal product (M-COP, Fig. A1) includes the cloud optical thickness (M-COT), cloud effective radius (M-REF), cloud top
temperature/height/pressure (M-CTT, M-CTH, M-CTP), and the cloud water path (M-CWP) for daylight conditions. During
nighttime, only the cloud top temperature/height/pressure can be provided. The instrumental characteristics of the MSI sensor,
which largely determine the design and accuracy of the M-COP retrievals, are described in more detail in Wehr et al. (2023).
Within the planned processing chain, M-COP relies on the availability of cloud mask and cloud phase information as provided
also by the M-CLD processor in the first step, which is described in Hünerbein et al. (2022).

The paper is structured as follows: In section 2, the operational Level 2 M-COP algorithms will be described in detail.
Section 3 presents some examples of the verification of the M-COP products. This has been performed by applying the M-
CLD processor to the EarthCARE simulator test scenes (Donovan et al., 2023), the MODerate-resolution Imaging Spectrometer
(MODIS) measurements and Spinning Enhanced Visible and Infrared Imager (SEVIRI) measurements. Finally, results will be
summarized and discussed in section 4.

## 2   M-COP algorithm description

The M-COP retrieval uses multiple MSI channels in visible/shortwave infrared and thermal range of the electromagnetic
spectrum to derive cloud optical thickness (M-COT), effective cloud radius (M-REF) and multiple estimates of cloud height
(M-CTT, M-CTH, M-CTP) as output products. The algorithm is embedded in the M-CLD processor framework. While it is
in many aspects similar to already existing cloud retrievals for current sensors in space (e.g., MODIS, SEVIRI), it has been
implemented from scratch following a mathematically optimal estimation approach. An overview of the workflow is given in
Figure 1. The main idea is to compare simulated reflectance and radiance values for a wide range of possible atmospheric
conditions with the measured values from the sensor. The simulations, which build the forward model of the retrieval, are
partly created in advance with a radiative transfer model (RTM) and then stored in look-up-tables. We decided to use an
optimal estimation (OE) inversion technique (OE, (Rodgers, 2000), sec.2.4), because this provides very nicely an uncertainty
propagation from the measurement through the forward model to a product uncertainty estimates. M-COP assumes cloud
mask and cloud phase as a known input. These parameters are supplied from directly preceding procedures within the M-
CLD processor (Hünerbein et al., 2022). Furthermore, background information of atmospheric and surface state, such as



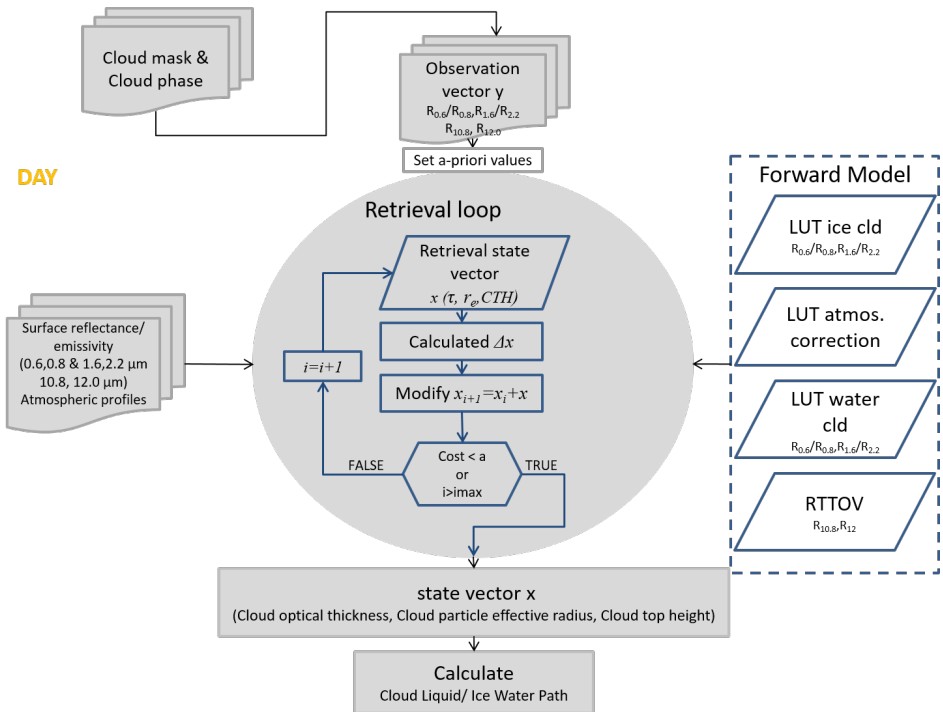

**Figure 1.** Flowchart of the M-COP algorithm. The algorithm is applied to all cloudy MSI pixels.

humidity and temperature profiles and surface reflectance and emissivity, is needed. We get this information from the auxiliary meteorological database (X-MET, Eisinger et al. (2023)), which is specifically initiated for this mission. The OE technique finds the best estimate of the cloud properties by an iterative retrieval loop, which compares the forward model result of an

assumed state vector with the observation by taking into account the respective uncertainties. Once both, the forward model output of the assumed state vector, and the observation vector, are close enough to meet specific requirements (this means then it minimizes a cost function), the retrieval was successful, and the state vector represents the output. The OE can also include a-priori knowledge of the to-derived products. We present in the following subsections details of the components of M-COP.

## 2.1 Input data

The MSI L1c products are calibrated and geolocated instrument data. The measurements for solar channels are given in radiance and the three terrestrial channels in brightness temperature. The L1c product is re-sampled onto the grid of a reference TIR channel (Eisinger et al., 2023). Therefore, each channel is mapped on the same pixel, which is important for the M-CLD processor, where we use the combination of all channels. The algorithm starts with the calculation of the reflectances at the top of the atmosphere in the solar channels. The reflectances ( $R(\theta_0, \theta, \phi)$ ) of each channel $i$ are obtained from the following input

parameters, the measured radiance ($L$) and the corresponding inband solar irradiance $E_0, i$ as

$$R(\theta_0, \theta, \phi) = \frac{\pi L_i(\theta_0, \theta, \phi)}{E_{0,i} cos(\theta_0)} , i = VIS, NIR, SWIR1, SWIR2.$$





The measured radiance and brightness temperature for each pixel depends on the solar zenith angle $\theta_0$, the viewing zenith angle $\theta$ and the relative azimuth angle $\phi$, defined by the difference between sun and instrument viewing azimuth angles.

## 2.2 Ancillary data

The retrieval requires further background information on atmospheric and surface conditions. The forward model includes a reflectance term which accounts for backscattered radiation from the underlying surface. For land pixels, we use the long-term albedo climatology from the MODIS science team (Moody et al., 2008). We assume that the white-sky albedo product is best suited for cloudy atmospheres, since it refers to the reflection of incoming diffuse radiation. MODIS has very similar channel specifications so that these data can be used as it is. Over ocean, the surface albedo is assumed to be 0.05 at both 0.6 $\mu$m and 115 1.6 $\mu$m. For the forward model in the longwave channels, we use the land surface emissivity values provided by the MODIS land surface temperature and emissivity products, which provides per-grid temperature and emissivity values (Wan (2014)).

## 2.3 Forward model

The forward model is an operator which simulates the observed radiance at the sensor based on a known atmospheric state and given auxiliary input parameters. It is strictly channel based, so that the complete operator is composed of independent 120 simulations of the channel i from a set of:

$$F_i(x) = y$$

with x is the state vector of the atmosphere (including the cloud properties) and y the vector of the observations (the measurements). The forward operator F includes the auxiliary data with their uncertainties. The individual components of the equation 125 are separated into the solar spectrum with negligible terrestrial emission (up to 3.5 micron wavelength), and the part of the spectrum where thermal emission dominates the radiative transfer.

### 2.3.1 Visible and shortwave infrared radiative transfer

This part of the forward model is built on an existing retrieval algorithm, the Cloud Physical Properties developed at KNMI (CPP, Cloud Physical Properties retrieval, Roebeling et al. (2006)), which has been used operationally for SEVIRI and AVHRR 130 for more a decade. It has been validated and improved over the years (Greuell and Roebeling (2009), Roebeling et al. (2013), Benas et al. (2017)).

Radiative transfer in this range is formulated in terms of reflectance values. The measured reflectance $R_{toa}$ at the top of the atmosphere can be expressed for a known surface reflectance $\alpha_s$ , and a known geometrical constellation as

$$R_{toa}(\tau, R_e, \theta_0, \theta, \phi) = R_{cl}(\tau, r_e, \theta_0, \theta, \phi) + \frac{\alpha_s t_c(\tau, r_e, \theta_0) t_c(\tau, r_e, \theta)}{1 - \alpha_s \alpha_a(\tau, R_e)} \qquad (1)$$

where $\tau$ is the cloud optical thickness, $r_e$ the effective radius of cloud particle distribution, $\alpha$ is the surface reflectance, $t_c$ cloud transmittance, which is needed for both angles, incoming $\theta_0$ and outgoing $\theta$, $R_{cl}$ the cloud reflectance, and $\alpha_a$ is the



hemispherical sky albedo for upwelling, isotropic radiation also known as spherical albedo. Other effects below and above the cloud layer are neglected in this equation. The target is to find the pair of $\tau, r_e$ which gives the highest accordance, or better the optimal estimate for the set of equations above for all VIS/NIR channels. Within the retrieval loop, a pair of $\tau, r_e$ is assumed

initially (the a-priori), and the result $R_{toa}$ of the forward model is compared to the observation to adjust the assumption till an optimal result is found. Thus, the functions $R_{cl}$ $\theta_c$ and $\alpha_a$ have to be simulated during the retrieval loop. Since this is not a simple and fast task, it is carried out in advance, and the results are stored in look-up-tables (LUT), and being interpolated from there to speed up the processing.

To generate these tables we use the radiative transfer model, DAK (Doubling-Adding KNMI, de Haan et al. (1987), Stammes

(2001)). To take in account also the Rayleigh scattering and ozone absorption above a cloud, the simulations were carried out with standard atmospheric states. Table 1 summarizes the LUT settings together with binning of individual dimensions. DAK can simulate cloud reflectance, cloud transmittance and cloud spherical albedo without the influence of surface reflectivity, so that the second term on the right side of Equation 1 takes this into account directly during the optimization process.

The DAK calculations concern monochromatic radiative transfer at a wavelength close to the centre of the respective satellite

channel. These calculations neglect scattering and absorption by atmospheric gases, except for Rayleigh scattering by air molecules and absorption by ozone.

The atmospheric correction for scattering and absorption processed in layers above cloud top is derived by MODTRAN radiative transfer model Berk et al. (2000). The atmosphere corrected top of atmosphere reflectance ($R_{atm.corr.}$) is calculated as:

$$R_{atm.corr.} = R_{toa}t_{a,ac}(\theta_0, z_{ct}, H)t_{a,ac}(\theta, z_{ct}, H, TCO) \tag{2}$$

where $t_{a,ac}$ is the above-cloud atmospheric transmission simulated by MODTRAN using a Lambertian surface placed at the cloud top height ($z_{ct}$, M-CTH) and for a given water vapor path ($H$, humidity profile) and total column ozone ($TCO$). The two-way transmission, i.e., the product of the two transmission values, is a function of the geometrical air mass factor ($AMF = 1/_0 + 1/$). This two-way transmission is calculated in advance for a wide range of AMFs and stored in a LUT

with dimensions AMF, $z_{ct}$, and $H$. Absorption by trace gases within and below the cloud is neglected. More details on the implementation of atmospheric correction and the effect on retrieved cloud properties can be found in (Meirink et al., 2009).

### 2.3.2 Terrestrial radiative transfer

In contrast to the VIS/NIR/SWIR1/SWIR2 forward model the thermal part includes mainly the emission from surface, atmosphere and cloud layers. The calculations are done with RTTOV 11.3 Saunders et al. (2009, 1999). The forward simulations of

the infrared radiances are used to compare the observed MSI brightness temperature with the simulated brightness temperature to retrieve the M-CTT based on the method of Fritz and Winston (1962). The simulations require atmospheric profiles e.g. temperature ($T$) and humidity ($H$), which are provided by the X-MET product Eisinger et al. (2023). The top of the atmosphere radiance ($L$) or brightness temperature ($T_b$) at $8.8\mu m$, $10.8\mu m$ and $12.0\mu m$ will be simulated once for clear sky and black cloud (emissivity equal 1) condition. The overcast radiance ($L$) consists of contributions from four terms, transmission





**Table 1.** Summary of the setting to generate the M-COP LUTs

| Parameter | Settings | |
|---|---|---|
| Vertical profiles of pressure, Temperature, and ozone | Midlatitude summer | |
| Aerosol model | None | |
| Solar zenith angle $(\theta_0)^a$ | 0-84.3°] (73 Gaussian points in $\mu_0 = cos(\theta_0)$) | |
| Viewing zenith angle $(\theta)^a$ | Same as $\theta_0$ | |
| Relative azimuth angle $(\phi)^a$ | 0-180°] (equidistant, 91 points) | |
| Cloud optical thickness $(\tau)$ | 0-256 (equidistant in $log(\tau)$, 22 points) | |
| | water clouds | ice clouds |
| Cloud particle types | Spherical water droplet | General habit ice crystal[b] |
| Cloud particle sizes | 3-34$\mu$m (equidistant in $log(r_e)$) | 5-80$\mu$m (equidistant in $log(r_e)$) |
| Complex refractive index | Segelstein(1981) | Warren and Brandt (2008) |

a) The chosen distributions of agles are motivated in Wolters et al. (2006).

b) Baum et al. 2014

radiance upwelling from below cloud level ($L_s$ ), emission from the cloud ($L_c$), reflection of radiance downwelling from above the cloud level and emission of radiance from the atmosphere above the cloud ($L_{at}$).

$$L(x,\theta,\epsilon) = L_s + L_c + L_{at} \qquad (3)$$

The surface partition can be formulated with:

$$L_s(x,\theta,\epsilon_s) = \epsilon_s B(T_s)t_a(\theta)t_c(\theta) \qquad (4)$$

and is a function of surface emissivity $\epsilon_s$ and the Planck function of surface temperature. Both values are assumed as known from the X-MET data set. $t_c = 1 - \epsilon_c$; for optically very thick clouds the cloud emissivity is equal to 1. In that case the surface as well as the atmospheric layer below the cloud is not contributing to the TOA radiance. $t_a$ is the transmission of the atmosphere. The cloud part of Equation 3 is formulated as

$$L_c(x,\theta,\epsilon_c) = \epsilon_c(\tau,\theta)B(T_c)t_{ac}(\theta) \qquad (5)$$

where we again have to assume the emissivity of the cloud layer, and the cloud top temperature. The cloud emissivity can be parameterised by $\epsilon_c = 1 - exp[-\tau_{ir}/cos\theta]$ by assuming the optical thickness in IR range with $\tau_{ir} = 0.5\tau_{vis}$ ((Minnis et al., 1993))

The value $t_a$ depicts the entire atmospheric transmission from cloud top to the top of the atmosphere. The radiance coming from atmospheric layers outside the cloud is simulated with RTTOV 11.3 for different atmospheric levels $z$ based on auxiliary





data $(T, P, H)$ profiles.

$$L_{at}(\theta, \epsilon_s) = \int\limits_{t_s}^{1} B(T_a)dt + [1 - \epsilon_s] * t_s(\theta)^2 \int\limits_{t_s}^{1} \frac{B(T_a)}{t(\theta)^2}dt \tag{6}$$

whereby $t_s$ is the total transmission from surface to the top of the atmosphere.

Note that we can modulate the cloud emissivity by changing the cloud optical thickness and the cloud top height as an input

parameter.

## 2.4  Retrieval Method

The retrieval loop (Figure 1) uses 1D-var optimal estimation inversion techniques to retrieve the cloud optical thickness, effective radius and cloud top heights during daytime, which is defined by the solar zenith angle ($\theta < 84°$). At nighttime the retrieval relies on the infrared channels and only the cloud top height is retrieved.

### 2.4.1  Optimal estimation

The basic principle of the M-COP algorithm follows the description of Rodgers (2000). The OE method is commonly used to solve the inverse problem (e.g., Watts et al. (2011), Walther and Heidinger (2012), Poulsen et al. (2012)). The OE does not provide an explicit solution but a class of solutions and assigns a probability density to each. The probability $P(x|y)$ of the retrieval state for all possible solutions $x$ for a given observation $y$ is defined as: $P(x|y) = \dfrac{P(y|x)P(x)}{P(y)}$

– $P(x)$: prior PDF of the atmospheric state $x$

– $P(y|x)$: conditional PDF of $y$ given $x$; requires knowledge of forward model and statistical description of the measurement error

– $P(y)$: prior PDF of the measurement $y$

The solution is found by maximising the probability $P(x|y)$ related to the values of the state vector $x$. The a priori estimate

of the state is defined by $x_a$ and the measurement vector $y$ is defined as a three element vector with visible, shortwave IR and infrared radiances.

$$\boldsymbol{x} = \begin{pmatrix} \tau \\ r_e \\ CTT \end{pmatrix} \text{ and } \boldsymbol{y} = \begin{pmatrix} R_{vis} \\ R_{swir} \\ T_b \end{pmatrix}$$

The state is mapped into the measurement space with the forward model $F(x)$ (see section 2.3.1, 2.3.2). The OE based on the assumption that the errors in the measurements $S_e$ and background $S_a$ can be described by a Gaussian distribution. Therefor

the state vector $x$ should be preferable also Gaussian distributed. To account for this the $\tau$ and $r_e$ are transformed in the logarithmic space with a base two for the inversion, which leads also to a more Gaussian error distribution. In order to find the



solution an iterative search is done by maximize $P(x|y)$ or minimize the cost function $J$, which comprises the measurement error and the background knowledge:

$$J(x) = (F(x) - y)S_e^{-1}(F(x) - y)^T + (x - x_a)S_a^{-1}(x - x_a)^T)$$

The M-COP retrieval loop starts with a defined a-priori values of the state vector $x$ and the observation $y$. The cost $J$ is calculated for each iteration step. Each iteration step of the retrieval loop requires search events in the forward model operators $F$. The derived radiances by the forward model $F(x_i)$ are compared to measurement, $y$ which defines the cost function. The optimal estimation searches for the minima in the cost function until convergence. The Levenberg-Marquardt descent method is used for the minimization within an iteration process (Marquardt (1963); Levenberg (1944)). If the cost reaches a pre-defined

threshold, the solution is found and the retrieval loop will end.

### 2.4.2 Measurement vector and covariance matrix

A global set of SEVIRI data has studied in order to establish typical top of the atmosphere radiances/brightness temperatures. The SEVIRI instrument aboard MSG (Meteosat Second Generation) offers excellent measurements to study the global variation

of the radiances. SEVIRI covers the visible (VIS006, VIS008), the near infrared (NIR016), and the thermal infrared spectral region (IR039, IR062, IR073, IR087, IR097, IR108, IR120, IR134) with twelve channels. These SEVIRI channels match the spectral position of the same MSI channels. This gives the possibility to use the SEVIRI data set for comparison of the maximum and minimum radiances, as well as for the specification of the typical radiance. To get the full range of the four seasons the criteria for the data set were: time: 9 UTC; day: in the mid of the months; month: March, June, September,

December; year: 2007; for all SEVIRI pixels.

The histogram distribution for cloudy, clear and mixed cases have been studied (see Fig. 2) as well as the maximum and minimum radiances/brightness temperatures have been taken from cloud free and cloudy atmosphere. From the mixed cases the typical scene radiances were estimated from the 50th percentile of the data. The typical radiances of the MSI channels are

specified with the comparable Seviri channels (see Tab. 2).

The measurement and background knowledge are described by the covariance matrices. The measurement error $S_e$ is defined through the instrument signal-to-noise ratio or the noise equivalent different temperature (Wehr et al., 2023) and the forward model error as:

$$S_e = (\frac{L_{type}}{SNR})^2 + \sum (K_{acc})^2$$

The typical radiances $L_{type}$ are taken from the above described SEVIRI analysis. The forward model errors are calculated by the Jacobians $K$ and given geophysical variable error $acc$ for each iteration step. For the background covariance, $S_a$ the variance of the state vector is used. The uncertainties are assumed to be independent of each other so that the off-diagonal elements can be set to zero.





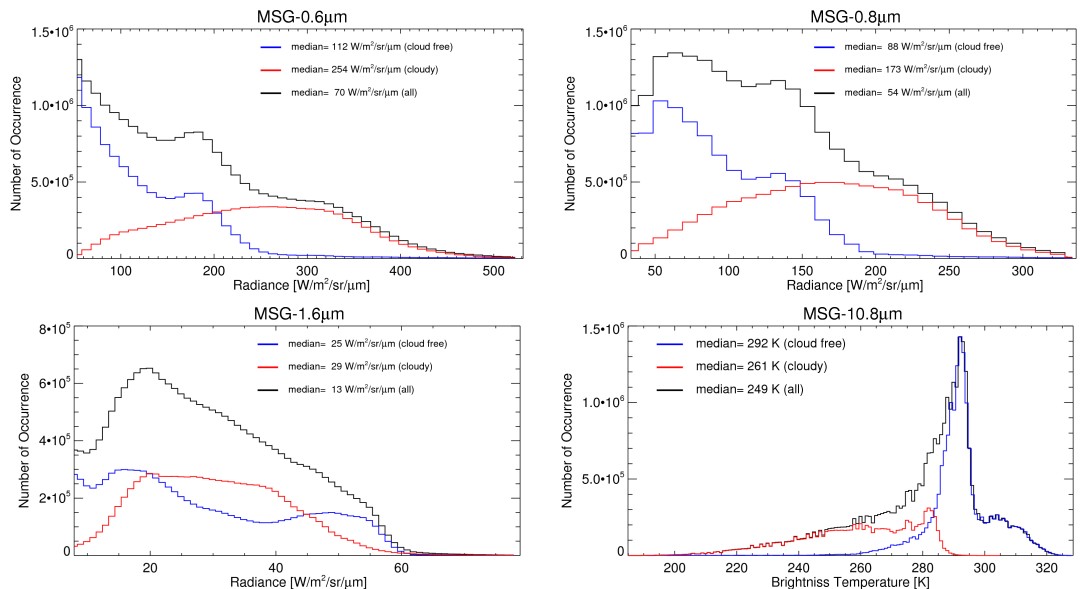

**Figure 2.** Histogram of the radiance in VIS, NIR, SWIR-1 and the brightness temperature TIR-2 based on SEVIRI measurements

**Table 2.** Comparison of the minimum and maximum radiances/brightness temperatures from the SEVIRI channels.

|  | VIS | NIR | SWIR-1 | SWIR-2 | TIR-1 | TIR-2 | TIR-3 |
|---|---|---|---|---|---|---|---|
| $L_{min}[W/m^2/sr/\mu m$ or $K]$ | 53 | 34 | 8 | 0.8 | 185 | 185 | 185 |
| $L_{max}[W/m^2/sr/\mu m$ or $K]$ | 531 | 336 | 79 | 338 | 328 | 328 | 328 |
| $L_{typ}[W/m^2/sr/\mu m$ or $K]$ | 54 | 40 | 9 | 4 | 286 | 286 | 285 |

## 2.5 Cloud optical and physical properties

The derived Level-2 cloud properties from the OE retrieval schema are M-COT, M-REF and M-CTT. The M-CTT is further converted to M-CTH and M-CTP using the atmospheric profiles (X-MET) of pressure, temperature and height. These properties are accompanied by uncertainty measures derived from the OE algorithm. The Figure 3 presents the full suite of M-COP products for one EarthCARE test scene, which refers to a realistic EarthCARE frame and MSI swath.

## 2.6 Cloud water path

The relation between the cloud optical thickness and the cloud effective radius is used to calculate the cloud water path based on the assumption of a homogeneous cloud. The cloud water path is calculated only indirectly due to the fact that the measurements from the MSI have no water absorption channel. The cloud water path $CWP$ is defined as the integral of the liquid/ice water content throughout the profile of an ice/water cloud layer. MSI measurements of solar reflectance in shortwave atmospheric window channels are used to retrieve the cloud phase (water or ice on the top of the cloud, (Hünerbein et al.,



2022)), cloud optical thickness ($\tau$) and the effective radius ($r_w$) or ice particle radius ($r_i$). The cloud liquid/ice water path is calculated as a function of $\tau$ and $r_{w,i}$. The CWP is derived by using the following equation Stephens (1984):

$CWP = \frac{2}{3}\tau r_{w,i}\rho_{w,i}$

where $\rho_w$ is the density of liquid water 1 $g/cm^3$ and $\rho_i$ is the density of ice 0.916 $g/cm^3$. The CWP error is estimated from the combination of the error of $\tau$ and $r_{w,i}$ as follows:

$\Delta CWP = \tau\Delta r_{w,i} + r_{w,i}\Delta\tau$

## 3  Performance and intercomparison / algorithm results

### 3.1  Evaluation with synthetic test scene

The M-COP algorithm performance has been tested by applying the M-CLD processor to different atmospheric test scenes
created with the EarthCARE simulator. This is an end-to-end simulator for the EarthCARE mission capable to simulate the four instruments configurations for complex realistic scenes. Specific test scenes have been created from model output data, see (Donovan et al., 2023). The 6000 km long frames include different types of clouds and aerosols as well as surface and illumination conditions. These synthetic scenes make it possible to evaluate and to intercompare the different cloud properties from active and passive sensors as e.g., cloud liquid water path or the cloud effective radius (Mason et al., 2022). The synthetic
test scenes set up have been also considered as the truth to better understand and quantify the differences between the retrieved cloud properties based on the different measurement principle. It should be noted that the simulated test scenes are intended to quantify the performance of the different processors, but not to tune the retrievals because the test scenes strongly depend on the assumption made in the EarthCARE simulator. In the following, we present results obtained with the M-CLD processor for the HALIFAX scene (Donovan et al. (2023),van Zadelhoff et al. (2022)). The scene starts with clouds over the Greenland
ice sheet followed by high backscatter/extinction clouds down to 50° N. A high ice cloud regime starting over east Canada down to 35° N is followed by a low-level cumulus cloud regime imbedded in a marine aerosol layer below an elevated dirty dust later around 5 km altitude. The results for the M-COP products for the HALIFAX scene are presented in Fig.3 with the corresponding error. The M-COT error generally increases with increasing M-COT above 50, which is expected has the measurement at the non absorption channel (VIS) get saturated with increasing COT. The M-REF error show clear separation
between the detected cloud phase, where ice cloud associate with higher errors. Note above 60°N the solar zenith angle is higher than 84° due to nighttime condition. Therefore, M-COT/M-REF cannot be retrieved, only the M-CTH is retrieved for this region.

In this evaluation, the 3d dataset input for the test scene (called as the truth) is used to define the presence of clouds and to calculate the modelled cloud optical thickness. This is calculated from the extinction profiles of all hydrometeors multiplied
by the layer thickness (Figure 5a) and compared to the retrieved M-COT. The true cloud optical thickness can also be retrieved in nighttime condition. Further, the truth includes values below 0.01, which are generally classified as clear sky by the M-







**Figure 3.** M-COP for the HALIFAX scene with: M-COT (top, left), error of M-COT (top, right), M-REF (middle, left), error of M-REF (middle, right), CTT (bottom, left), error of CTT (bottom, right)




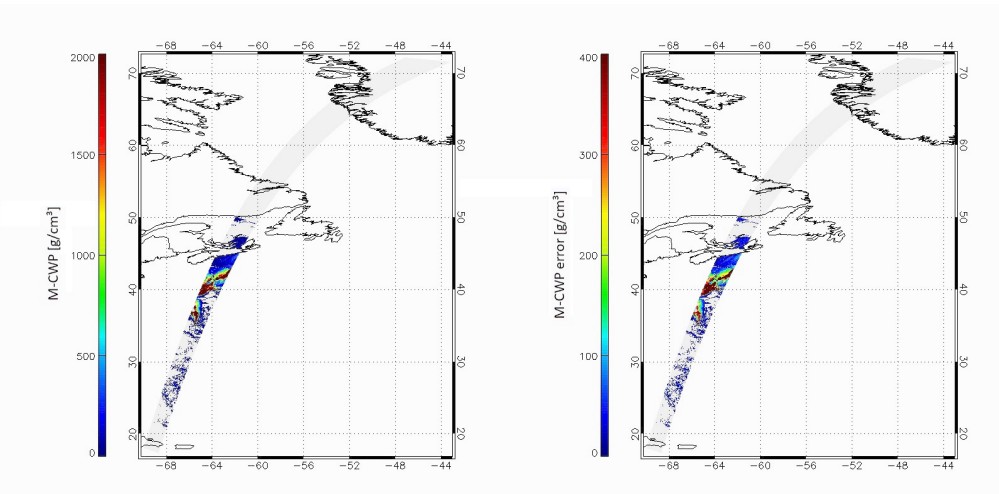

**Figure 4.** M-CWP of the HALIFAX scene (left) and the error of M-CWP (right)

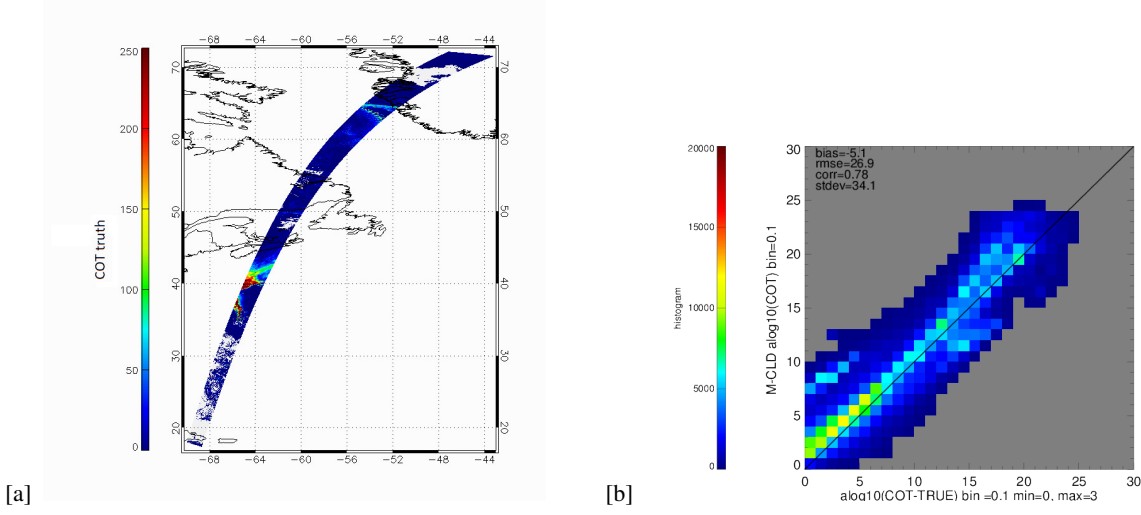

**Figure 5.** Cloud optical thickness for the Halifax scene a) true cloud optical thickness at 680nm and b) Histogram of the true COT and M-COT

CM cloud mask. This has been discussed in Hünerbein et al. (2022). Overall, the comparison with the retrieved cloud optical thickness M-COT (Figure 5b) shows a good and promising agreement (Figure 5).

## 3.2 MODIS data for testing M-COP

Additionally, to the synthetic test scenes, the M-COP cloud algorithm has been also tested against realistic satellite observation from MODIS. For that purpose, MODIS Terra L1b calibrated radiances have been used as input for the M-CLD processor.



The MODIS channel 1, 2, 6, 7, 29, 31 and 32 are closed to the seven MSI channels and taken to replace the MSI signal. The meteorological auxiliary data required as input for the M-CLD processor are replaced by CAMS (Copernicus Atmosphere Monitoring Service) forecast data field. The cloud mask and phase (M-CM) are produced within the M-CLD processor, and the results are described in the companion paper Hünerbein et al. (2022). The LUTs are not specifically adapted to the MODIS filter function, which will cause some uncertainties. The Figure 6 shows a case study over the Capo Verde islands in the Atlantic Ocean. The right column shows the M-CTT, M-COT and M-REF results as retrieved by the M-CLD processor. The corresponding MODIS L2 products are given at the left column. The CTT comparisons show an underestimation of the cloud top height for high cirrus clouds, especially multilayer clouds, which will be further investigated if an improvement is possible by using multilayer flags. The COT comparison of M-CLD and MODIS exhibits an almost perfect match, with a correlation factor of 0.95 and a bias of less than 1. The REF comparisons show more differences, such as for ice clouds the M-REF have smaller particle sizes but for water clouds the sizes are slightly higher. Compared to MODIS the MSI cloud product shows in general a good agreement. The robustness of these results could be demonstrated by using longer time periods and also other regions, which is however not shown in this study.

## 3.3 Intercomparison with other SEVIRI cloud products

The M-CLD cloud products have also been validated within the framework of the CGMS International Cloud Working Group (ICWG, Roebeling et al. (2013), (Roebeling et al., 2015)). The ICWG is a working group aimed at harmonizing and advancing quantitative cloud property retrievals. In support of its mission, the ICWG has assessed a large number of level-2 passive imager cloud parameter retrievals through inter-comparison activities. The ICWG cloud assessment have focused on quantifying deterministic differences between level-2 cloud parameters (and their error estimates), and the statistical validation against superior reference dataset. This assessment so far has been limited to geostationary satellites, i.e., Meteosat Second Generation (MSG) with the SEVIRI instrument. SEVIRI provides similar channel settings to MSI except for the 2.2$\mu$m channel, which is missing. The cloud parameters that have been part of the assessment are: cloud mask, cloud top temperature, cloud top pressure, cloud top height, cloud phase, cloud optical thickness, effective particle size, cloud liquid water path, and cloud ice water path. As reference datasets, the main sources of information have been the cloud properties obtained from CLOUDSAT and/or CALIPSO observations and Cloud Liquid Water Path observations from passive microwave instruments (e.g., AMSR). For a number of "golden days" different scientific institutions have contributed their cloud datasets for intercomparison among others the EUMETSAT central facility, the Nowcasting SAF and the Climate Monitoring SAF. The cloud properties retrieved from all MSG SEVIRI scans collected during - 13 June 2008, 19 August 2015 and 21 July 2016 had to be submitted and served as basis for the intercomparison (Hamann et al., 2014). It should be noticed that the MSI M-CLD algorithm is developed for a polar orbiting satellite and is adapted to the MSI specification, which lead to uncertainties in the adaptation to SEVIRI due to differences in central wavelength and spectral response function, radiative transfer simulations and generated look-up-tables. The M-CLD processor has been directly applied to the SEVIRI observation.

Figure 7 shows the cloud top pressure obtained by thirteen different retrievals for the noon scene of the 13th June 2008, the combined cloud top pressure and the RGB-composite together with the A-Train track. The cloud top pressure is separated





**Figure 6.** Case study from MODIS Terra granule at 1220UTC, 30 September 2021.The top row shows the cloud top temperature, the middle panel shows the cloud optical thickness and the bottom row shows the cloud effective radius at 1.6 $\mu$m, whereby the left column belongs to the MODIS L2 products and the right column to the M-CLD .)





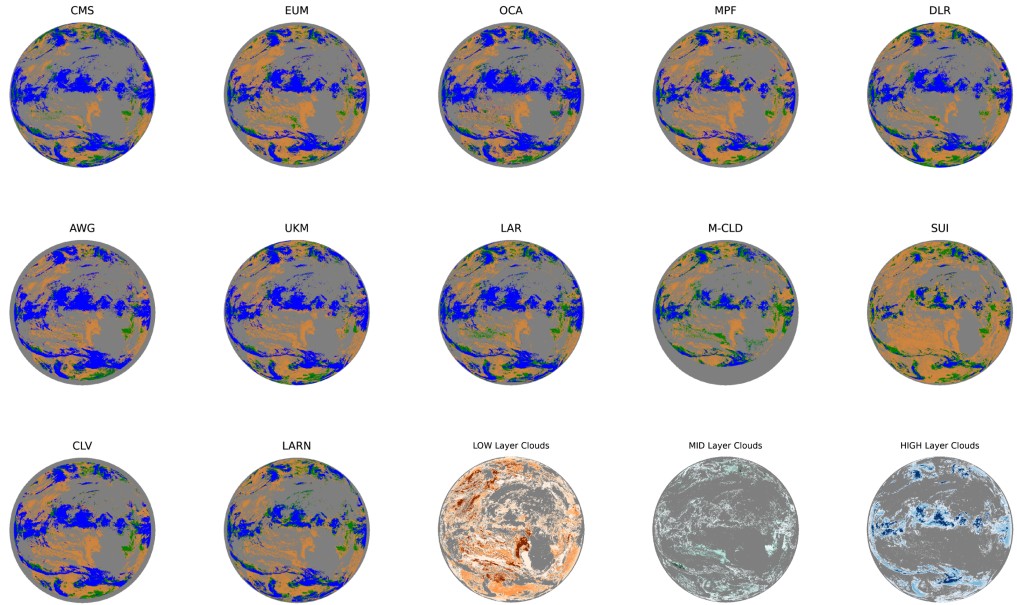

**Figure 7.** Cloud top pressure heights of the thirteen algorithms submitted to ICWG, classified into three cloud classes: low (orange), middle (green) and high (blue) based on the ISCCP classification: for 13-06-2008 at 12:00 UTC. The combined cloud top pressure classes based the consensus of all thirteen ICWG algorithms are shown in the bottom row, separated in low (third MSG disc), middle (fourth MSG disc) and high (fifth MSG disc) .

in three classes for the comparison: low (red), middle (green) and high (blue) based on the ISCCP classification. The zonal distribution of the cloud top pressure is comparable for all datasets. High clouds are present in the intertropical convergence zone. The combined cloud top pressure illustrates the disagreement for the anvil of the convective clouds within the intertropical convergence zone. Adjacent to them, low clouds are most common in the marine stratocumulus region between 30° S and 30°

N and agree well between the algorithms. Note that the cloud masks differ between the algorithms, which also influences the mean cloud top pressure obtained by the algorithm. Some algorithms also limit the domain for retrieval due to large viewing or solar zenith angles and/or sun glint.

    Figure 8 shows the histograms of the cloud top temperature from all the applied retrievals. A common cloud mask is used to calculate the individual CTT histograms. Different cloud occurrence frequencies are observed for the boundary layer clouds.

Most histograms show a distribution with two cloud occurrence maxima, one around 230 K and a second one below 280 K. The weakness of M-CTT is the detection of multilayer clouds with high cirrus clouds in the upper troposphere. For example, the anvil of the deep convective storm systems classified as with medium instead of high cloud top height (see Figure 7). This is also reflected in the histogram (Figure 8, yellow curve) where the second maximum at 230 K is not present. It should be noted that the other algorithms take advantage of the SEVIR $CO_2$ band, which MSI does not have.



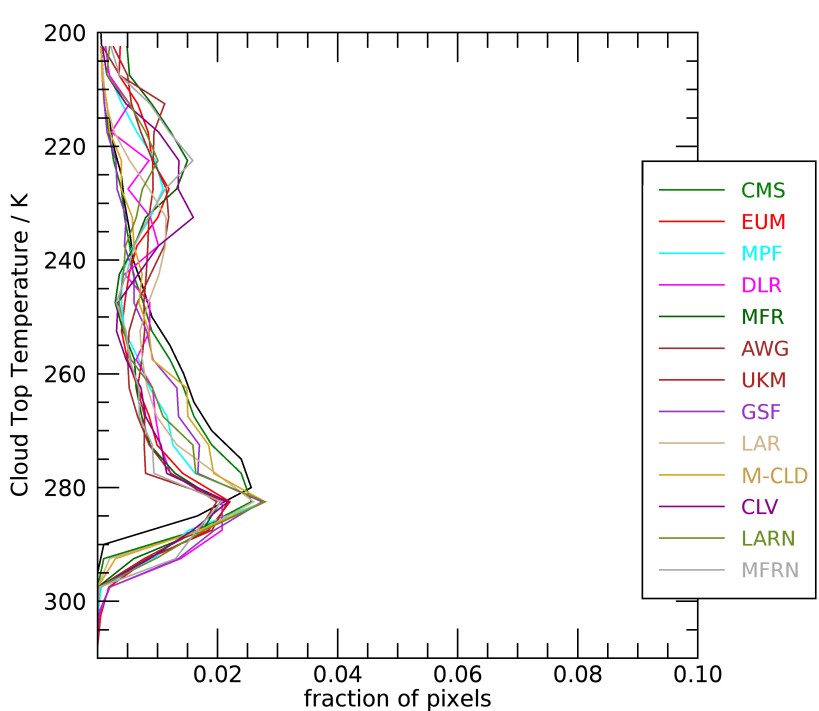

**Figure 8.** Histograms of the cloud top temperature of the thirteen ICWG algorithms for 13-06-2008 at 12:00 UTC based on a common cloud mask.

Additionally, the A-Train constellation of Earth-observing satellites (Stephens et al. (2002), Stephens et al. (2018)) are used to quantify the accuracy of the different cloud products. For the purpose of the comparison, the CloudSat and MODIS measurement are re-gridded onto the SEVIRI grid. Five regions from the SEVIRI disc have been selected to study in more detail specific meteorological conditions. One example is a deep convection cell over West Africa is presented in Figure 9. On top, the CloudSat measurements on the A-Train track are compared with the M-CTH (white crosses, Figure 9), which show good

agreement. The M-CLD COT and REF values (red: ice phase, green: water phase) are compared well to those from the MODIS instrument (blue, Figure 9).



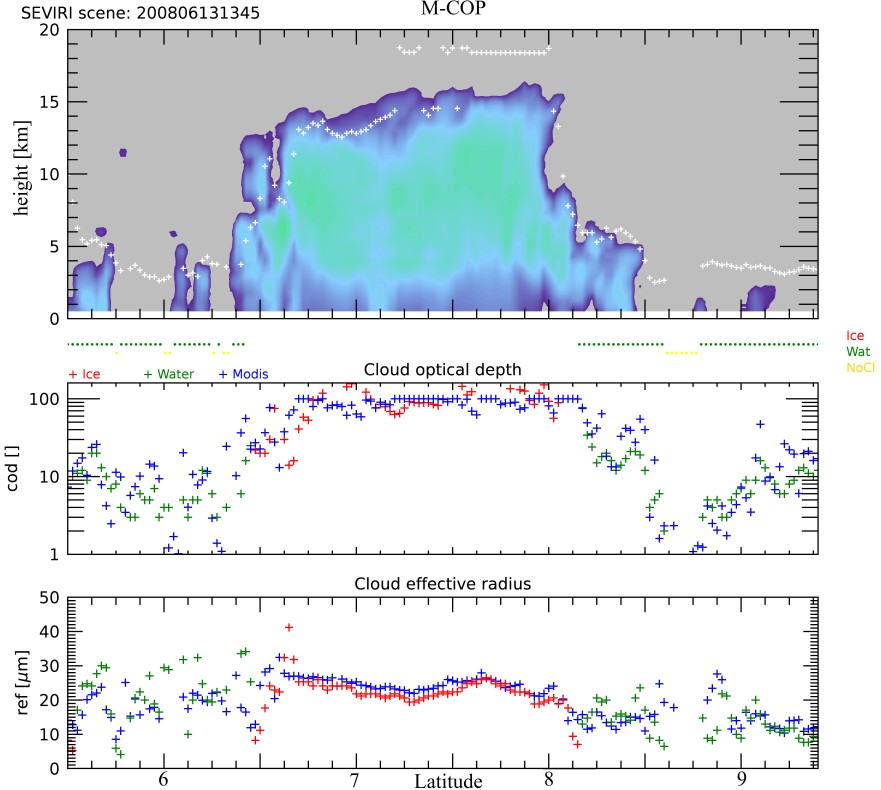

**Figure 9.** Selected CloudSAT, MODIS and M-COP (based on SEVIRI L1) cloud properties over West Africa with deep convection. Top panel, Cloud radar reflections are shown with the M-CTH (dotted white line). The middle panel gives on the track MODIS-COT(blue) and M-COT (with cloud phase red: ice, green: water) and the lower panel gives similar the cloud effective radius.

## 4 Conclusion

This paper describes the baseline algorithms used to retrieve the optical and physical products (M-COP) from observations of the MSI instrument onboard the EarthCARE satellite. As such, M-COP is an essential part of the MSI cloud processor M-CLD. The cloud mask algorithm M-CM, which is an important input for M-COP, is described in the accompanying paper Hünerbein et al. (2022). We described all significant components of the M-CLD processor to retrieve the M-COP products (Figure A1). The algorithm is based on the widely used optimal estimation technique and simultaneously considers the shortwave and longwave MSI channels to obtain an optimal estimate of the targeted cloud properties, including an error estimate. The validation results demonstrate the comparability of M-COP results with those of similar operationally used algorithms. M-CLD is able to provide cloud optical and physical products in near real time on a global basis. It has to be realized however that the performance depends on the observation conditions including surface type (desert, sun-glint, ice), illumination and viewing geometry, and cloud type. The provision of uncertainty estimates as part of the M-COP products will enable an assessment of



the situation-dependent retrieval uncertainty. The software will be embedded in the ESA processing framework.

The algorithm performance has been assessed using synthetic EarthCARE test scenes as well as satellite observations from the MODIS and SEVIRI instruments. Overall, the comparison with the test scene truth and the different imager cloud products show good and encouraging results. The MODIS and SEVIRI imagers feature additional channels which provide further information, e.g., for detecting thin clouds or identifying multilayer cloud situations (thin over thick). In particular, the cloud top height is observed to be biased low in multilayer situations, which are further investigated with the synergistic ATLID-MSI

retrieval in Haarig et al. (2023).

The MSI solar channels show a spectral curvature nonlinearity disturbance due to imperfect bandpass filters on the curved optical lenses (Wehr et al. (2023)), which causes a spectral shift in across-track direction. This degradation of the spectral information content is currently under investigation, and it is planned to integrate mitigation measures in the M-CLD processor. Further, the forward model RTTOV will be updated soon to its newest version 13.3, which includes the real MSI filter functions

as one important update. During the commissioning phase, the configurable parameter of the processor will be still adjusted and optimized to improve the accuracy by using dedicated EarthCARE campaigns and geostationary satellites observations.

*Data availability.* The EarthCARE Level-2 demonstration products from simulated scenes, including the MSI cloud mask products discussed in this paper, are available from https://doi.org/10.5281/zenodo.7117115 (van Zadelhoff et al., 2022).

*Author contributions.* The manuscript was prepared by AH, SB and HD. The M-COP code was developed by AH and JF. AH validated the

M-COP algorithm against MODIS, SEVIRI and simulated test scenes. GZ generated the CPP LUTs. AW generated the dataset and created the plots for the ICWG results.

*Competing interests.* The authors declare that they have no conflict of interest.

*Acknowledgements.* This work has been funded by ESA grants (IRMA), 4000112018/14/NL/CT (APRIL) and 4000134661/21/NL/AD (CARDINAL). We thank Tobias Wehr and Michael Eisinger for their continuous support over many years, and the EarthCARE developer

team for valuable discussions during various meetings.



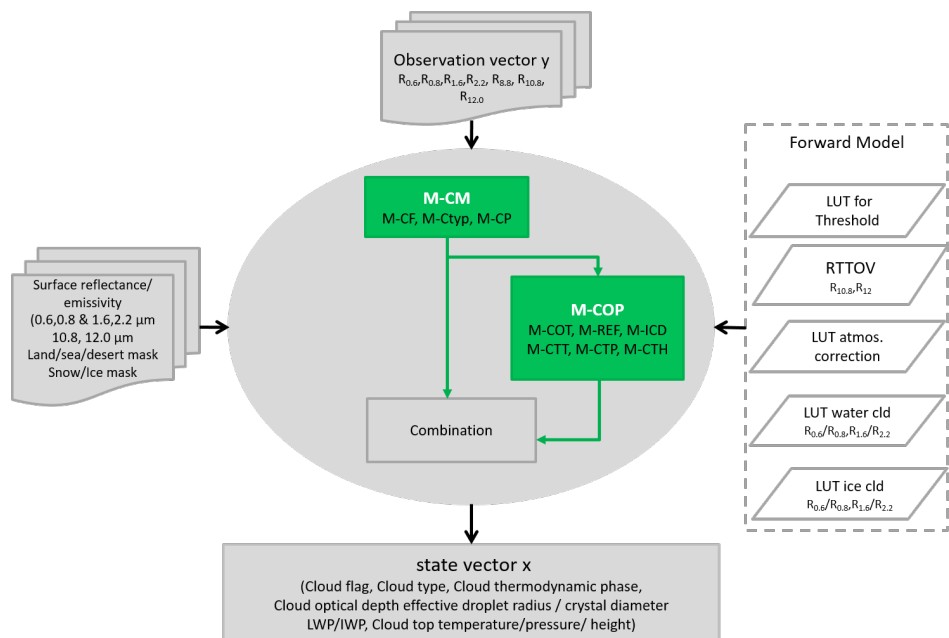

**Figure A1.** Flowchart of the M-CLD processor.

## Appendix A

The MSI cloud processor consists of two main parts (see Figure A1), which are sequentially processed. First the cloud mask (M-CM), which will be mandatory for the other cloud retrievals (M-COP).



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
