# Peer review of "Cloud optical and physical properties retrieval from EarthCARE multi-spectral imager: the M-COP products"

_EGUsphere, 2023_

## Referee Comment (RC1)

**Overview**

The manuscript provides an overview of the cloud optical/physical algorithm (M-COP) that will be used in processing EarthCARE MSI observations. Retrieved datasets include cloud top properties (temperature, pressure, height), optical thickness and effective radius; cloud water path is subsequently derived from the product of optical thickness and effective radius. The algorithm does not include cloud masking or thermodynamic phase, which are provided by a separate upstream MSI product that feeds into M-COP.

The text provides reasonably sufficient information for those in the cloud imager retrieval community to understand the basic algorithm methodology, radiative transfer modeling and ancillary data assumptions. The algorithm leverages a long heritage, i.e., the algorithm and/or methodology is not new. Therefore, there is low risk that the algorithm's datasets won't be of immediate use and interest to the community soon after the launch of EarthCARE.

While not a novel algorithm, the manuscript is certainly within the scope of AMT and its readership.

**Major Comments**

1. There are no references for the MSI instrument (or ATLID, CPR) or the EarthCARE mission.

2. Given that this is a heritage-based algorithm, the manuscript lacks sufficient up-to-date references to other similar datasets being produced from global imagers. A literature search is needed to provide context.

L35 paragraph: There are no references supporting "hence … several long-term … data records". Presumably the authors are relying on the preceding references that provided a brief historical perspective but these aren't up to date or complete. There are no reference for the latest/recent products from MODIS (GSFC group, CERES group, European and Japanese groups?), VIIRS (NOAA DCOMP/NCPOMP, NASA CLDPROP), next generation GEO imagers (NOAA ACHA, JMA?, CM-SAF?) imager cloud products.

A related example: a MODIS cloud product is mentioned on L298 without any reference or a statement of data provenance (product name, version, archive site). Similarly, details on MODIS Terra L1B are missing (reference, product name – will indicate spatial resolution used, version, archive). By the way, the text says that Terra radiances are being used but, for the shortwave, the authors should be reading in the reflectance dataset (based on the on-board solar diffuser, the primary calibration); see related comment below under L106.

3. CloudSat/A-Train

L46, L47 sentences are misleadingly negative about constellation flying. Remove the "Drawback …" and later text unless a case is made for why it's relevant to the algorithm and this manuscript.

L46: "susceptible to temporal changes in atmospheric state". When the A-Train satellites were fully functional, the two active sensors were constellation flying within ~15 sec of each other and MODIS Aqua was ahead of them by ~60 sec. What reference is there that suggests constellation flying with these temporal differences compromised science due to atmospheric state changes?

L47: "While it was possible …". Doesn't make sense as a follow-on to the preceding sentence. Regarding colocation: "impossible to produce the synergistic CloudSat, CALIPSO and MODIS product with sufficient accuracy". The reference Kato et al. 2010 does not state this. It's unclear what CloudSat operations issue is being referring to (no reference) or when it occurred. If it refers to anomalies that happened after the Kato reference (e.g., CloudSat DO-OP mode), this simply points out that hardware anomalies can have consequences, just as they will on EarthCARE.

By the way, an advantage of constellation flying is that the active sensor flight tracks can be offset from the imager track so as to better avoid imager sun glint contamination.

4. Sect. 2.4.2:

L240: *Ltype* supposed to be *Ltyp* (as in L typical)?
L241: subscript *acc* not defined.

I didn't understand the point of this section or what I should learn from Fig. 2 and Table 2.

The authors didn't connect the measurement variability to the measurement error Se. *Ltyp* is said to be taken from SEVIRI measurements but where did SNR come from? I had assumed SNR was from a pre-launch laboratory calibration (or perhaps on-orbit by looking at dark/cold scenes) and so can only be used to predict noise if it's the denominator for the same *Ltyp* that was used in the SNR calculation.

If the authors can justify keeping the figure/table, shortwave channel radiances should be changed to reflectances so it will have meaning to the readers. The authors converted IR radiance to BT in Fig. 5 for that very reason but still used radiance units in the table.

**Minor**

• Table 1:

  o None of the references cited in the table are in the reference list.
  o To what extent is the SZA/VZA and azimuthal resolution capable of resolving cloud glory and cloudbow features in the backscattered reflectance? Does it matter?
  o Add information about the cloud particle size distribution/effective variance being used.
  o What spectral bands are the cloud LUTs calculated for? The text is vague, seems to use VIS generically to also mean NIR (0.865) for optical thickness information (L61), and then shows an effective radius retrieval using the 1.65 µm channel (Fig. 6) while not specifying what channel is used in other figures.
  o Remove two occurrences of "]"
  o "a)" footnote misspells angle

• Abstract, 1st sentence: "that" instead of "which".

• Abstract: Regarding OE and uncertainties, the wording in L88 reads fine. But it's incorrect to infer in the abstract that OE has an advantage (at least not uniquely) over other solution methods in enabling full error propagation. Ignoring prior uncertainties (not a practical constraint in these type of cloud

retrieval problems), the retrieval uncertainty covariance matrix can be calculated independent of the methodology (e.g., Platnick et al., 2021, *Rem. Sens.*, Eq. 1).

• L17, 18: I didn't understand why that part of the sentence was there. The paragraph speaks to a separate rationale for having cloud retrieval products and could just start with "There is an urgent …"

• L38: "their measurement's principle" – awkward if not grammatically incorrect. Could try "information content" though information is used later in the sentence.

• L47, 48: CALIPSO is uppercase.

• L56: "-35" meaning minus with respect to what? Specify EarthCARE's MLT.

• L61: absorbing is for SWIR (not NIR) using notation from a couple sentences previous.

• L70 paragraph: This is a good place to mention the products that are produced daytime only. Also, worth pointing out here that the cloud mask/phase is provided as input to M-COP.

• L76: The MODIS acronym has already been used. Regardless, definitions not provided in the text for most satellite acronyms.

• L95: "Once both …" re-write. Not grammatically correct with position of commas and "both".

• L97: "was" should be "is"

• L106: Does the algorithm actually use radiance and not reflectance for the shortwave optical retrievals? I would have though MSI would have an onboard solar diffuser that will be used for the primary VNIR/SWIR channel calibration and therefore eliminate the need for E0 spectral irradiance database.

• L112: The Moody et al. is getting a bit old and doesn't account for regional ecosystem changes (either interannual variability or secular).

• L149: DAK calculations are at a single wavelength at the channel center. How broad are the MSI channels? Does a single channel center calculation correctly model cloud radiative properties (especially in the SWIR)?

• L199 and L214 equations aren't necessary. Plenty of references/historic literature with that detail.

• L256+: the author's own notation for effective radius (re) is not being used in this equation and in-line notation. Can remove the "w,i" subscript from r (otherwise optical thickness would also have that notation as it's a joint retrieval with effective radius).

• L273: Does the simulator uses different cloud particle scattering models for ice than M-COP? Different radiative transfer code?

• Fig. 3, L247, 278:

In the figure caption and text, does "error" refer to the OE retrieval uncertainty (if so, that's what it should be called) or is it the difference between the retrieval and an absolute truth? I think it's the former but not sure. Please clarify.

The CTT contrast is not enough to infer an obvious cloud phase. Please add a phase plot (could be added to Fig 4 or Fig 5 where there's room). I realize that phase is not your algorithm but it's very hard to interpret effective radius retrievals without it.

The caption should explain that the COT and REF retrieval/error are missing in that region because the sun angle is too low. However, L280 says that happens at 60N and it appears that the shortwave retrieval data goes missing at 50N. Please clarify.

Why is CTT error (retrieval uncertainty?) missing north of the Canadian mainland?

• Fig 5a: Shouldn't the image be masked off at the day/night boundary like the previous optical retrieval images?

• Fig. 6: The ice phase radius differences may be due to ice model LUT differences. What can you say about the size difference expected between the Baum model (Table 1) and whichever MODIS cloud product is being used in the comparison (reminder that the reference is missing)? How does the 2.25 μm channel size compare between the two retrievals? Please answer both in the text. L295: by the way, why isn't the M-COP algorithm being run by recalculating LUTs for MODIS spectral channels so the comparisons can be more meaningful? Apparently, that uncertainty is a source of concern.

• Fig 7: I don't see the value in showing every dataset in this figure. Also, the dataset labels are not the same list as in the Fig. 8 legend. Why?

• Fig. 8: Where's the M-CTT line?

• Fig. 9: What MODIS COT and REF product is shown? Again, no reference. The labels are "optical depth" when they should be "optical thickness" for consistency throughout the text.

---

## Author Comment (AC1)

**Reply to Anonymous Referee #1**

We would like to thank Anonymous Referee #1 for their helpful comments. Below are the original comments in regular with our responses in *italic* text.

**Comments to the author**:

The manuscript provides an overview of the cloud optical/physical algorithm (M-COP) that will be used in processing EarthCARE MSI observations. Retrieved datasets include cloud top properties (temperature, pressure, height), optical thickness and effective radius; cloud water path is subsequently derived from the product of optical thickness and effective radius. The algorithm does not include cloud masking or thermodynamic phase, which are provided by a separate upstream MSI product that feeds into M-COP.

The text provides reasonably sufficient information for those in the cloud imager retrieval community to understand the basic algorithm methodology, radiative transfer modeling and ancillary data assumptions. The algorithm leverages a long heritage, i.e., the algorithm and/or methodology is not new. Therefore, there is low risk that the algorithm's datasets won't be of immediate use and interest to the community soon after the launch of EarthCARE.

While not a novel algorithm, the manuscript is certainly within the scope of AMT and its readership.

**Major Comments**

1. There are no references for the MSI instrument (or ATLID, CPR) or the EarthCARE mission.

*Thanks, the reference (Wehr et al. 2023) was given two sentences later, but we agree we should provide this already earlier. Now we have added the reference directly after the first-time mention EarthCARE.*

2. Given that this is a heritage-based algorithm, the manuscript lacks sufficient up-to-date references to other similar datasets being produced from global imagers. A literature search is needed to provide context.

L35 paragraph: There are no references supporting "hence … several long-term … data records". Presumably the authors are relying on the preceding references that provided a brief historical perspective but these aren't up to date or complete. There are no reference for the latest/recent products from MODIS (GSFC group, CERES group, European and Japanese groups?), VIIRS (NOAA DCOMP/NCPOMP, NASA CLDPROP), next generation GEO imagers (NOAA ACHA, JMA?, CM-SAF?) imager cloud products.

A related example: a MODIS cloud product is mentioned on L298 without any reference or a statement of data provenance (product name, version, archive site). Similarly, details on MODIS Terra L1B are missing (reference, product name – will indicate spatial resolution used, version, archive). By the way,  the text says that Terra radiances are being used but, for the shortwave, the authors should be reading in the reflectance

dataset (based on the on-board solar diffuser, the primary calibration); see related comment below under L106.

*We have added several more recent publications for AVHRR, MODIS, VIIRS, AHI and SEVIRI around line 35.*

*"A variety of methods and data sets exist for the determination of cloud microphysical properties from multi-spectral satellite imagery, for example from AVHRR (Derrien et al. (1993); Kriebel et al. (1989); Nakajima and King (1990); Walther and Heidinger (2012)), ATSR (Along Track Scanning Radiometer) (Watts et al., 1998), MODIS (King et al., 1997; Platnick et al., 2016), VIIRS (Platnick et al., 2021), AHI (Advanced Himawari Imager, Letu et al., 2020), the GOES series (Minnis and Harrison, 1984; Heidinger et al.,30 2020) and SEVIRI (Spinning Enhanced Visible and InfraRed Imager, Stengel et al., 2014). All these methods share a fundamental principle: the reflection of clouds in the visible wavelength region, specifically in a non-absorbing channel, is primarily determined by cloud optical thickness. In contrast, the reflection function at the absorbing channels in the shortwave-infrared region primarily relies on cloud particle size."*

*Reference to MODIS data. We changed/added in the chapter 3.2:*

*"MODIS Terra L1b calibrated radiance" to "MODIS Terra L1b calibrated reflectance"*

*Line 298: ".. The corresponding MODIS L2 products (MOD06_L2, Platnick et al. (2015b)) ..."*

*We included also ".. M-REF (at 1.6µ m) .." in the text to be more clear that we used not the MODIS standard REF , but the comparable one to the MSI REF.*

*In chapter 3.3 we added:*

*"Additionally, the A-Train constellation of Earth-observing satellites (Stephens et al., 2002, 2018) of the comparison, the CloudSat (Marchand et al., 2008) and MODIS measurement (MYD06_L2, Platnick et al. (2015a)) are re-gridded onto the SEVIRI grid."*

*Further we added to "Data availability" paragraph the following sentence:*

*".. ). MODIS Level-1 (MODIS Science Data Support Team (SDST), 2017) and the MODIS Level-2 cloud product MOD06 (Platnick et al., 2015a) and MYD06 (Platnick et al., 2015b) are available through the Level-1 and Atmosphere Archive Distribution System Distributed Active Archive Center (LAADS DAAC: https://ladsweb.modaps.eosdis.nasa.gov, last access 31-01-2023). The Cloudsat 2B-GEOPROF product (Marchand et al., 2008) can be ordered via (CloudSat data processing center: https//www.cloudsat.circa.colostate.edu). Thanks are also given to EUMETSAT for providing the MSG2-SEVIRI data.).."*

3. CloudSat/A-Train

L46, L47 sentences are misleadingly negative about constellation flying. Remove the "Drawback …" and later text unless a case is made for why it's relevant to the algorithm and this manuscript.

*Our intention was not to sound negative in any way. We remove the sentences.*

L46: "susceptible to temporal changes in atmospheric state". When the A-Train satellites were fully functional, the two active sensors were constellation flying within ~15 sec of each other and MODIS Aqua was ahead of them by ~60 sec. What reference is there that suggests constellation flying with these temporal differences compromised science due to atmospheric state changes?

L47: "While it was possible …". Doesn't make sense as a follow-on to the preceding sentence. Regarding colocation: "impossible to produce the synergistic CloudSat, CALIPSO and MODIS product with sufficient accuracy". The reference Kato et al. 2010 does not state this. It's unclear what CloudSat operations issue is being referring to (no reference) or when it occurred. If it refers to anomalies that happened after the Kato reference (e.g., CloudSat DO-OP mode), this simply points out that hardware anomalies can have consequences, just as they will on EarthCARE.

By the way, an advantage of constellation flying is that the active sensor flight tracks can be offset from the imager track so as to better avoid imager sun glint contamination.
4. Sect. 2.4.2:

*We agree with this statement and had remove the sentences.*

L240: Ltype supposed to be Ltyp (as in L typical)?
*Yes, done.*

L241: subscript acc not defined.
*We give more explanation to it and added a table with the acc_n (geophysical variable error).*
*"The forward model errors are calculated by the Jacobians K and the geophysical variable error acc_n for each iteration step. The applied geophysical variable errors are provided in Tab. 3. "*

I didn't understand the point of this section or what I should learn from Fig. 2 and Table 2.

The authors didn't connect the measurement variability to the measurement error Se. Ltyp is said to be taken from SEVIRI measurements but where did SNR come from? I had assumed SNR was from a prelaunch laboratory calibration (or perhaps

on-orbit by looking at dark/cold scenes) and so can only be used to predict noise if it's the denominator for the same Ltyp that was used in the SNR calculation.

If the authors can justify keeping the figure/table, shortwave channel radiances should be changed to reflectances so it will have meaning to the readers. The authors converted IR radiance to BT in Fig. 5 for that very reason but still used radiance units in the table.

*We see your point. When we started the development of the retrieval, we got only an assumed SNR. To start we wanted to use realistic values. Therefor, we used SEVIR data to get an idea. But you are right this will be adapted during the commission phase. The uncertainty of the radiometric accuracy will be measure with the on-board calibration diffusers. Also, a MSI error pixel values are planned to have in the MSI-RGR, which we could use as well for the measurement error. In the moment it is still in discussion and that why we could not add it.*

*We added following sentence. "..are taken from the above described SEVIRI analysis. This will be adapted during the commissioning phase to on-board measured uncertainties and radiances."*

*In Tab2 we correct the units. As the SNR values are refer to radiances Wehr et al. 2022, we kept radiances to be consistent.*

**Minor**

• Table 1:

  o None of the references cited in the table are in the reference list.
  *Added.*
  o To what extent is the SZA/VZA and azimuthal resolution capable of resolving cloud glory and cloudbow features in the backscattered reflectance? Does it matter?
  *We did not analyse this topic so far. Therefore, we cannot answer this question.*

  o Add information about the cloud particle size distribution/effective variance being used.

  *We used for the water clouds the two-parameter gamma size distribution with an effective variance of 0.15. We added this to the table.*

  o What spectral bands are the cloud LUTs calculated for? The text is vague, seems to use VIS generically to also mean NIR (0.865) for optical thickness information (L61), and then shows an effective radius retrieval using the 1.65 µm channel (Fig. 6) while not specifying what channel is used in other figures.

*We added in the table capture a sentence to be clearer. "Table 1. Summary of the setup for generating the M-COP LUTs for VIS and SWIR1 channels."*

o Remove two occurrences of "]"
*Done.*

o "a)" footnote misspells angle
*Done.*

• Abstract, 1st sentence: "that" instead of "which"
*Done.*

• Abstract: Regarding OE and uncertainties, the wording in L88 reads fine. But it's incorrect to infer in the abstract that OE has an advantage (at least not uniquely) over other solution methods in enabling full error propagation. Ignoring prior uncertainties (not a practical constraint in these type of cloud retrieval problems), the retrieval uncertainty covariance matrix can be calculated independent of the methodology (e.g., Platnick et al., 2021, Rem. Sens., Eq. 1).

*Thanks, the abstract has been adjusted to: "…The algorithm is based on the solar and terrestrial MSI channels within an optimal estimation framework. This framework enables full error propagation given by the uncertainties in measurements and a-priori information."*

• L17, 18: I didn't understand why that part of the sentence was there. The paragraph speaks to a separate rationale for having cloud retrieval products and could just start with "There is an urgent …"
*Thanks, we have taken your suggestion and removed the first part of the sentence.*

• L38: "their measurement's principle" – awkward if not grammatically incorrect. Could try "information content" though information is used later in the sentence.
*Thanks, we have taken your suggestion.*

• L47, 48: CALIPSO is uppercase.
*Done.*

• L56: "-35" meaning minus with respect to what? Specify EarthCARE's MLT.
*We changed the sentence to: "MSI will have a swath width of 150 km, asymmetrically tilted away from the sun and covering 35 km to right side and 115 km to the left side of nadir. "*
*The EarthCARE's mean local time is planned to be after Wehr et al.: "The MRD requires an equator crossing time between 13:15 and 14:00 hours." As no fixed time given in the paper, we can not specify the MLT of EarthCARE.*

• L61: absorbing is for SWIR (not NIR) using notation from a couple sentences previous.

*Thanks, we correct this.*

• L70 paragraph: This is a good place to mention the products that are produced daytime only. Also, worth pointing out here that the cloud mask/phase is provided as input to M-COP.

*We changed the order and hope it get clearer: "… MSI cloud processor (M-CLD, Fig. A1). The M-CLD consists of two main parts, which are sequentially processed. First the cloud mask (M-CM) is processed with the output of cloud flag (M-CF) and cloud phase (M-CP) product (Hünerbein et al., 2023), which are mandatory to retrieve the cloud optical and physical product (M-COP). The M-COP includes the cloud optical thickness (M-COT), cloud effective radius (M-REF), cloud top …"*

• L76: The MODIS acronym has already been used. Regardless, definitions not provided in the text for most satellite acronyms.

*We remove the MODIS definition on this place and add it to the first time, where we have used it. We also add the missing satellite acronyms for VIIRS …*

• L95: "Once both …" re-write. Not grammatically correct with position of commas and "both".

*We rephrased the sentence to:*
*"Once the forward model output of the assumed state vector, and the observation vector satisfies the requirement of the minimization of a cost function, the retrieval process is considered successful. This state vector represents then the solution."*

• L97: "was" should be "is"

*Done.*

• L106: Does the algorithm actually use radiance and not reflectance for the shortwave optical retrievals? I would have though MSI would have an onboard solar diffuser that will be used for the primary VNIR/SWIR channel calibration and therefore eliminate the need for E0 spectral irradiance database.

*Even so that MSI have an onboard solar diffuser. The MSI L1c product provide only spectral radiances and the solar spectral irradiance. Therefore, in the first step reflectance are calculated (chapter 2.1).*

• L112: The Moody et al. is getting a bit old and doesn't account for regional ecosystem changes (either interannual variability or secular).

*That is true. We used it in the first place, but we hope we can update this in a later state.*

• L149: DAK calculations are at a single wavelength at the channel center. How broad are the MSI channels? Does a single channel center calculation correctly model cloud radiative properties (especially in the SWIR)?

*We assume that the channels are spectrally narrow enough (VIS- 0.02μm, SWIR1-0.05μm) that cloud reflectance simulations at the central wavelength are a good approximation of the cloud reflectance over the channel. However, it remains an approximation and we did not evaluate how accurate it really is for the MSI channels. We propose to add a sentence to the manuscript: "… of the respective satellite channel. This implies an assumption that reflectance at the central wavelength is close enough to the average reflectance over the channel. The radiative transfer calculations neglect scattering and absorption …'"*

• L199 and L214 equations aren't necessary. Plenty of references/historic literature with that detail.

*We would like to keep the equation even it is known as structure of the text is based on it.*

• L256+: the author's own notation for effective radius (re) is not being used in this equation and in-line notation. Can remove the "w,i" subscript from r (otherwise optical thickness would also have that notation as it's a joint retrieval with effective radius).

*Thanks, corrected to $r_e$.*

• L273: Does the simulator uses different cloud particle scattering models for ice than M-COP? Different radiative transfer code?

*Yes. M-COP used the General habit model from Baum et al. 2014. The EarthCARE End-to End Simulator the phase function from ice and snow are adapted from Baum et al. 2014, but for cloud ice and snow the aggregated solid columns properties were used (detailed description given in Donovan et al. 2022), which are different.*
*Further the used radiative transfer code for the MSI forward model in the EarthCARE End-to End Simulator frame based on DISORT (discrete ordinate algorithm, Stamnes et al. 1998) as the LUT for M-COP used DAK.*

• Fig. 3, L247, 278:
In the figure caption and text, does "error" refer to the OE retrieval uncertainty (if so, that's what it should be called) or is it the difference between the retrieval and an absolute truth? I think it's the former but not sure. Please clarify. The CTT contrast is not enough to infer an obvious cloud phase. Please add a phase plot (could be added to Fig 4 or Fig 5 where there's room). I realize that phase is not your algorithm but it's very hard to interpret effective radius retrievals without it.

*The error refers to the OE retrieval and we follow your suggestion and call it now uncertainty. In Fig 5. we added the cloud phase and refer to it in the text.*

The caption should explain that the COT and REF retrieval/error are missing in that region because the sun angle is too low. However, L280 says that happens at 60N and it appears that the shortwave retrieval data goes missing at 50N. Please clarify. Why is CTT error (retrieval uncertainty?) missing north of the Canadian mainland?

*Thanks, we correct this to 50N.*
*We added: "The simple M-CTH night time retrieval provides no uncertainty values."*

• Fig 5a: Shouldn't the image be masked off at the day/night boundary like the previous optical retrieval images?
*Done.*

• Fig. 6: The ice phase radius differences may be due to ice model LUT differences. What can you say about the size difference expected between the Baum model (Table 1) and whichever MODIS cloud product is being used in the comparison (reminder that the reference is missing)?

How does the 2.25 μm channel size compare between the two retrievals? Please answer both in the text. L295: by the way, why isn't the M-COP algorithm being run by recalculating LUTs for MODIS spectral channels so the comparisons can be more meaningful? Apparently, that uncertainty is a source of concern.

*Yes, we agree the ice phase radius differences may be due to the different ice model LUT. We choose the MODIS, MOD06_L2 REF at 1.6μm as M-REF also retrieved at 1.6μm. We have to investigate this further more with real data.*

*We mention all MODIS channels which we need for the whole processor (M-CLD), which are imprecise as for the M-COP we need only 3 MODIS channels. We are not using the 2.25μm channel and we also did not use the standard effective radius from MODIS MOD06_L2. We have chosen the effective radius_1.6 to be comparable. "We added the sentence. "The three MODIS channels 1, 6 and 31 are used for the M-COP retrieval."*
*We added at the figure capture, which cloud product used and in section 3.2 and data availability the reference: MODIS L2 products (MOD06_L2, Platnick et al. (2015b)).*

• Fig 7: I don't see the value in showing every dataset in this figure. Also, the dataset labels are not the same list as in the Fig. 8 legend. Why?

*We disagree and do think that showing the three different products provides different arguments. While COT agrees very well between both, the MODIS L2 and the MSI M-CLD using the three MODIS channels, the CTT is too warm for M-CLD because it misses some of the thin cirrus clouds. Also, the Reff is underestimated by M-CLD which cannot be stated from just showing COT alone. Fig. 8 has a completely different*

*purpose, showing intercomparison results within the ICWG. Because these are the first verification of the M-CLD products with other retrievals, we think that the variety of these plots gives some added value.*

• Fig. 8: Where's the M-CTT line?
*We added in the caption of the figure: "The M-CTT product is named here under the processor name M-CLD."*

• Fig. 9: What MODIS COT and REF product is shown? Again, no reference. The labels are "optical depth" when they should be "optical thickness" for consistency throughout the text.
*We followed the suggestion and changed cod to COT and the title to Cloud Optical Thickness and added as well the reference.*

---

## Author Comment (AC2)

**Reply to Anonymous Referee #2**

We would like to thank Anonymous Referee #2 for their helpful comments. Below are the original comments in regular with our responses in *italic* text.

**Comments to the author**:

The manuscript describes an algorithm to derive cloud optical thickness, particles effective radius, and cloud top effective temperature using EarthCARE's multi-spectral imager (MSI). Cloud water path is also derived using optical thickness and effective radius. The retrieval uses 7 channels of MSI. The retrieval is done though a forward model in an iterative way to minimize a const function. The const function is the sum of the difference of modeled radiances and observed radiances and retrieval cloud properties and their a priori estimates. The const function takes into account of measurement uncertainty based on signal-to-noise ratio and forward model errors. The forward model errors are assumed to be independent. The algorithm is tested with EarthCARE simulator HALIFAX scene and MODIS data. Also, M-CLD properties are validated within the framework of the CGMS internatonal cloud working group.

The manuscript is well written and results are easy to understand. I only have minor comments or clarifications.

Equation 1. Is this for all 0.67, 0.865, 1.65, and 2.21 channels?

*Yes, principle yes, but for the LUT we calculate mainly the pair VIS/SWIR1. We added a sentence below the equation: The target is to find the pair of $\tau$, $r_e$ which gives the highest accordance, or better the optimal estimate for the set of equations above for the V IS/SW IR1 channels.*

Line 141. What is theta_c?

*Thanks, that is a mistake it should be t_c (cloud transmission) has been corrected.*

Line 242. S changes weights to sum up the elements to compute the cost function. How do you estimate diagonal term of Sa? Does this covariance matrix depend on region or cloud type? If variances are fixed, could you present the size of variances in a table? This is nice, in principle, if we know the covariance matrix. But in practice, we do not really know Sa. How sensitive the final solution is to the covariance matrix?

*For the background covariance, Sa we started with a simple assumption, which should be improve further. So far, the variance of the state vector is used based on the minimum and maximum values. We added a table (Tab.4).*

Table 1. Change "cloud particle size" to "cloud particle radius".

*Done.*

Figures. Generally, axis labels and legends are too small. For example, Figure 2 (legend), Figures 3, 4, 5 and 6 (labels for the color bar).

*We revised the figures.*

Figure 7 and 8 do not mean very much to readers who are not a member of the cloud working group unless all title or legends are explained.

*We think that it is important to show these figures in that level of detail, because the ICWG is a strong group with high expertise in cloud retrievals from passive satellite sensors and therefore it is a strong argument that the M-CLD retrieval can keep up with the other sophisticated retrievals in our opinion. The reference Hamann et al. 2014 provides much details about the retrievals and the intercomparison, which is out of the scope of the present manuscript, but it did not compare to M-CLD before. This was done just in a later step and never published. Therefore, we would like to include the results in our paper. We have added a sentence to the caption: Detailed information about the algorithms behind the acronyms in the titles are provided in table 4 of Hamann et al. 2014.*

Figure 9. Could you provide mean difference and RMS difference between MODIS M-COP values in a table? It is even better If the authors have compared more scenes and provide robust statistics.

*We agree that more statistics would certainly improve the significance of the comparison, but for the time being, we only have a few cases, for which statistical mean numbers should only be interpreted with care. However, once real data is available, there will be a comprehensive validation effort comparing to other sensors including the active instrument from EarthCARE. Fig. 9 (in the revised version now Fig. 10) shall give some qualitative indications about the agreement and maybe also weaknesses of the M-CLD retrieval, which should be further assessed with robust statistics in the future. We have slightly rephrased the text below the last figure: "The COT values between MODIS and MSI are in good agreement, while there is a slight overestimation of M-REF compared to MODIS for water clouds and underestimation for ice clouds. However, one should note that this is only a qualitative comparison for one case study and a comprehensive assessment of the quality of the M-CLD products will be perform once real MSI observations are available."*

---

## Referee Report (RR1)

• L30: The year for Platnick et al. (2016) is 2017 according to
https://ieeexplore.ieee.org/document/7707459.

• L33: Suggest slight editing for the following (green highlights are edited text):

"fundamental principle: the reflection of clouds in the visible/near-infrared wavelength region,
specifically in a non-absorbing channel, is primarily determined by cloud optical thickness. In contrast,
the reflection function at  absorbing channels in the shortwave and midwave-infrared region
 strongly depends on cloud particle size."

Note that SWIR/MWIF not necessarily "primarily" depending on REF, e.g., optically thin clouds (and
depending on spectral channel).

• In regard to Table 1, I previously asked "To what extent is the SZA/VZA and azimuthal resolution
capable of resolving cloud glory and cloudbow features in the backscattered reflectance? Does it
matter?" The response was: "We did not analyze … cannot answer the question."

Then please state in the manuscript that the algorithm's angular resolution has not been evaluated for
its ability to resolve liquid water cloudbow and glory phase function features in the reflected signal.

• L53

From my previous review, the authors response was: "We changed the sentence to: "MSI will have a
swath width of 150 km, asymmetrically tilted away from the sun and covering 35 km to right side and
115 km to the left side of nadir. …. no fixed time given in the paper, we can not specify the MLT of
EarthCARE."

Either use "east" and "west" instead right/left, or define how right/left is defined. And according to
Wehr et al. 2023 (https://doi.org/10.5194/amt-16-3581-2023) the MLT is known (1400 LT descending,
see Sect. 6).

• L93

"Once the forward model output of the assumed state vector, and the observation vector satisfies the
requirement of the minimization of a cost function, the retrieval process is considered successful. This
state vector represents then the solution."

Try: "Once the forward model output of the assumed state vector and the observation vector satisfy the
requirement of the minimization of a cost function, the retrieval process is considered successful. This
state vector then represents the solution."

• In response to "Does the simulator uses different cloud particle scattering models for ice than M-
COP? Different radiative transfer code?" the author's reply:

"Yes. M-COP used the General habit model from Baum et al. 2014. The EarthCARE End- to End
Simulator the phase function from ice and snow are adapted from Baum et al. 2014, but for cloud ice

and snow the aggregated solid columns properties were used (detailed description given in Donovan et al. 2022), which are different. Further the used radiative transfer code for the MSI forward model in the EarthCARE End-to End Simulator frame based on DISORT (discrete ordinate algorithm, Stamnes et al. 1998) as the LUT for M-COP used DAK."

Even if it's assumed that DISORT and DAK provide identical reflectances, it's not correct to compare M-COP ice cloud retrievals with the simulation if they use different particle scattering assumptions. Do you agree? If so, please comment on this apple-to-orange comparison in the manuscript text. Note that this is similar to the comment about "Fig. 7" below.

• Fig. 7 (Fig. 6 in original manuscript): Author's reply:

"Yes, we agree the ice phase radius differences may be due to the different ice model LUT. We choose the MODIS, MOD06_L2 REF at 1.6µm as M-REF also retrieved at 1.6µm. We have to investigate this further more with real data. We mention all MODIS channels which we need for the whole processor (M-CLD), which are imprecise as for the M-COP we need only 3 MODIS channels. We are not using the 2.25µm channel and we also did not use the standard effective radius from MODIS MOD06_L2. We have chosen the effective radius_1.6 to be comparable. "We added the sentence. "The three MODIS channels 1, 6 and 31 are used for the M-COP retrieval." We added at the figure capture, which cloud product used and in section 3.2 and data availability the reference: MODIS L2 products (MOD06_L2, Platnick et al. (2015b))."

From what I can tell of the response, the manuscript doesn't tell the reader that the REF comparisons are for different ice particle scattering models. Please mention this in the text (e.g., after the "robustness" sentence).

• L340 etc.

For Platnick et al. 2015a and 2015b, the links in the reference gave a 404 error. It looks like it will work if you remove the first occurrence of "doi.org/" in both URLs. Are you sure you downloaded and analyzed Collection 6.0 files instead of the more recent Collection 6.1 data that completed reprocessing in 2017/2018?

---

## Author Response (AR2)

**Reply to Anonymous Referee #1**

We would like to thank Anonymous Referee #1 for their helpful comments. Below are the original comments in regular with our responses in *italic* text.

**Minor Comments on revised version**

L30: The year for Platnick et al. (2016) is 2017 according to
https://ieeexplore.ieee.org/document/7707459.

*Thanks, have been corrected.*

2. L33: Suggest slight editing for the following (green highlights are edited text):

"fundamental principle: the reflection of clouds in the visible/near-infrared wavelength region, specifically in a non-absorbing channel, is primarily determined by cloud optical thickness. In contrast, the reflection function at the absorbing channels in the shortwave and midwave-infrared region primarily relies strongly depends on cloud particle size."

*Thanks, we took your suggestion.*

3. L53: From my previous review, the authors response was: "We changed the sentence to: "MSI will have a swath width of 150 km, asymmetrically tilted away from the sun and covering 35 km to right side and 115 km to the left side of nadir. …. no fixed time given in the paper, we can not specify the MLT of EarthCARE."

Either use "east" and "west" instead right/left, or define how right/left is defined. And according to Wehr et al. 2023 (https://doi.org/10.5194/amt-16-3581-2023) the MLT is known (1400 LT descending, see Sect. 6).

*We changed right/left to west/east.*

4. L93: "Once the forward model output of the assumed state vector, and the observation vector satisfies the requirement of the minimization of a cost function, the retrieval process is considered successful. This state vector represents then the solution."

Try: "Once the forward model output of the assumed state vector and the observation vector satisfy the requirement of the minimization of a cost function, the retrieval process is considered successful. This state vector then represents the solution.

*Thanks, we took your suggestion.*

5. In response to "Does the simulator uses different cloud particle scattering models for ice than MCOP? Different radiative transfer code?" the author's reply:

"Yes. M-COP used the General habit model from Baum et al. 2014. The EarthCARE End-to End Simulator the phase function from ice and snow are adapted from Baum et al. 2014, but for cloud ice and snow the aggregated solid columns properties were used

(detailed description given in Donovan et al. 2022), which are different. Further the used radiative transfer code for the MSI forward model in the EarthCARE End-to End Simulator frame based on DISORT (discrete ordinate algorithm, Stamnes et al. 1998) as the LUT for M-COP used DAK." Even if it's assumed that DISORT and DAK provide identical reflectances, it's not correct to compare MCOP ice cloud retrievals with the simulation if they use different particle scattering assumptions. Do you agree? If so, please comment on this apple-to-orange comparison in the manuscript text. Note that this is similar to the comment about "Fig. 7" below.

• Fig. 7 (Fig. 6 in original manuscript): Author's reply:

"Yes, we agree the ice phase radius differences may be due to the different ice model LUT. We choose the MODIS, MOD06_L2 REF at 1.6µm as M-REF also retrieved at 1.6µm. We have to investigate this further more with real data. We mention all MODIS channels which we need for the whole processor (M-CLD), which are imprecise as for the M-COP we need only 3 MODIS channels. We are not using the 2.25µm channel and we also did not use the standard effective radius from MODIS MOD06_L2. We have chosen the effective radius_1.6 to be comparable. "We added the sentence. "The three MODIS channels 1, 6 and 31 are used for the M-COP retrieval." We added at the figure capture, which cloud product used and in section 3.2 and data availability the reference: MODIS L2 products (MOD06_L2, Platnick et al. (2015b))."

From what I can tell of the response, the manuscript doesn't tell the reader that the REF comparisons are for different ice particle scattering models. Please mention this in the text (e.g., after the "robustness" sentence).

*Yes, we agree it always the problem that the assumptions are different. We added a sentence. "It should be noted that different ice particle scattering models are used, which account for some of the variance."*

6. L340 etc. For Platnick et al. 2015a and 2015b, the links in the reference gave a 404 error. It looks like it will work if you remove the first occurrence of "doi.org/" in both URLs. Are you sure you downloaded and analyzed Collection 6.0 files instead of the more recent Collection 6.1 data that completed reprocessing in 2017/2018?

*We correct the *bib entry. We had to add "/" before the _ from 06_L2 . Yes, we used collection 6.1 of MOD06L2. Therefor we add the citation Platnick et al. 2017 and collection 6.1 in the text.*